

# Fuzzysplit: demultiplexing and trimming sequenced DNA with a declarative language

Daniel Liu[1,2]

[1] Torrey Pines High School, San Diego, CA, USA
[2] Department of Psychiatry, University of California, San Diego, La Jolla, CA, USA

## ABSTRACT

Next-generation sequencing technologies create large, multiplexed DNA sequences that require preprocessing before any further analysis. Part of this preprocessing includes demultiplexing and trimming sequences. Although there are many existing tools that can handle these preprocessing steps, they cannot be easily extended to new sequence schematics when new pipelines are developed. We present Fuzzysplit, a tool that relies on a simple declarative language to describe the schematics of sequences, which makes it incredibly adaptable to different use cases. In this paper, we explain the matching algorithms behind Fuzzysplit and we provide a preliminary comparison of its performance with other well-established tools. Overall, we find that its matching accuracy is comparable to previous tools.

## INTRODUCTION

Advances in next-generation DNA sequencing technology allow large quantities of multiplexed DNA to be sequenced. Many methods, including the Genotyping by Sequencing (GBS) (*Elshire et al., 2011*) strategy, require sequenced DNA to first undergo preprocessing before further processing and analysis (e.g., identification of single nucleotide polymorphisms). The preprocessing step usually involves demultiplexing reads by barcodes/indexes and trimming adapters from reads.

The core of most preprocessing steps involve searching algorithms that find matching regions within the DNA sequence for some pattern sequence. The searching algorithm must also be fuzzy or approximate (i.e., allow mismatches or edits) to account for sequencing errors (*Kircher, Heyn & Kelso, 2011*). For example, when demultiplexing, reads of DNA are split into different files according to the barcode matched in the DNA sequence. Also, adapters in the DNA sequences need to be matched to identify the region to be trimmed. In the GBS method, restriction enzyme sites may also be matched. When handling paired-end reads, there may be combinatorial barcodes that require matching, for both the forward and the reversed reads.

Currently, there are many existing tools that implement a subset or all of those matching and trimming features above. Popular tools include FASTX-Toolkit (*Gordon & Hannon, 2010*), Axe (*Murray & Borevitz, 2018*), Flexbar (*Dodt et al., 2012*; *Roehr, Dieterich & Reinert, 2017*), Cutadapt (*Martin, 2011*), and AdapterRemoval (*Lindgreen, 2012*;

Corresponding author
Daniel Liu, daniel.liu02@gmail.com

*Schubert, Lindgreen & Orlando, 2016*), Trimmomatic (*Bolger, Lohse & Usadel, 2014*), Skewer (*Jiang et al., 2014*), and GBSX (*Herten et al., 2015*). However, they are mostly restricted to handling only a few types of DNA read schematics. With the development of new and more complex pipelines, the schematics of each read of DNA may change. For example, the aforementioned tools must be modified to handle out-of-read barcodes and unique molecular identifiers (*Kivioja et al., 2012*; *Islam et al., 2014*; *Girardot et al., 2016*). These modifications entail the development of more tools or features that can handle those specific formats of reads.

We aim to streamline the development of future pipelines by creating a general programming *language* that can easily handle different file formats and read schematics by describing them using a simple declarative language. The benefits of this approach is twofold. First, new schematics can be rapidly tested without first developing new tools or adapting previous tools to handle specific formats of DNA sequences. Second, by offloading the formatting of the DNA sequences to a special language, the source code of the tool itself does not need to scale in complexity in order to handle more DNA schematics and file formats (e.g., interleaved paired-end data, out-of-read barcodes, FASTQ formats).

In this paper, we introduce our idea of applying a domain-specific language (DSL) to match patterns in DNA sequences by discussing the algorithms necessary for matching arbitrary combinations of different patterns. We implement these algorithms in our Fuzzysplit tool, and we show empirical results for its accuracy and speed in comparison to other tools.

## FUZZYSPLIT

We present Fuzzysplit, a tool that can process a multitude of different input formats according to template files. The template files rely on a simple declarative DSL that defines the location and parameters of patterns to mirror the arrangement of barcodes, adapters, or other patterns that need to be matched. The templates are automatically repeated to handle multiple consecutive groups of lines within the input files.

Fuzzysplit's template files relies on three types of patterns that can be arranged to mirror the input text:

- **Fuzzy patterns.** These patterns are matched by measuring similarity between a piece of text and a subpattern (e.g., the reference barcode and the DNA sequence) through different edit distance metrics. Fuzzysplit supports Hamming distance, which allows substitution edits (*Hamming, 1950*), Levenshtein distance, which allows insertions, deletions, and substitutions (*Levenshtein, 1966*), and Damerau-Levenshtein distance, which is Levenshtein distance but with transpositions (*Damerau, 1964*). Fuzzysplit allows multiple fuzzy subpatterns (e.g., different barcodes) to be specified in a fuzzy pattern, but only one subpattern will be chosen as the matching subpattern. Fuzzysplit also supports wildcard characters like the undetermined N nucleotide.

- **Fixed-length wildcard patterns.** Each of these match a segment of text of a certain length if all characters in that segment are allowed by the pattern.

- **Interval-length wildcard patterns.** This matches a segment of text whose length lies within an interval and all characters in that segment are allowed by the pattern. Both

fixed-length and interval-length wildcard patterns are useful in matching sequences that contain repeating characters, like poly- A tails. Also, they can be used to match subsequences where the nucleotide can be anything, in order to cover the DNA area between the 5′ adapters/barcodes and 3′ adapters/barcodes.

Note that we distinguish the two wildcard patterns because they are implemented differently. Also, note that interval-length wildcard patterns are a generalization of fixed-length wildcard patterns. Fuzzysplit's patterns are based off of popular search tools like agrep (*Wu & Manber, 1992a*) that make use of symbols that describe patterns. In addition to those patterns, Fuzzysplit also supports:

- Defining variables that hold statistics, like length and subpattern index, for a matched pattern in the template files. The data of the matched pattern can be referenced by other patterns by using the variables. Also, these variables allow Fuzzysplit to output to different files depending on which patterns/subpatterns are matched. This is a generalization of demultiplexing.
- Allowing both optional and required patterns.
- Referencing data (e.g., barcodes, adapters, etc.,) that is stored in other list files from within the template files. Fuzzysplit also provides a selection operator that allows specific items to be selected from a list file based on some variable.
- Parallel processing of batches of input data through multithreading.

Overall, allowing arbitrary arrangements of patterns and match variables are the two core components of Fuzzysplit, as they allow Fuzzysplit to match a wide range of different formats. In addition to the arrangement of patterns, the template files also enable users to fine-tune exactly how each pattern is matched with pattern-specific parameters.

### Availability
Fuzzysplit is implemented in Java without any external libraries. Its compiled binaries and source codes are available here: https://github.com/Daniel-Liu-c0deb0t/Java-Fuzzy-Search. It is available under the MIT license.

## EXAMPLE
We show an example of how Fuzzysplit can be used through a demonstration that handles a simplified version of demultiplexing on a FASTQ file.

In a FASTQ file, each read is stored as four lines. It is important to note that changes to the second line, which contains the DNA sequence, must cause the same changes to the fourth line, which contains the quality scores for each nucleotide in the DNA sequence.

The following template file can be used to match 5′ barcodes in an input FASTQ file:

```
{i}
1 {f required, trim, name = "b", edits = 1, pattern = f"b.txt"}{i}
{i}
2 {r trim, length = %b.length%}{i}
```

In the template file, each pattern is denoted by {...}. The first character for each pattern indicates its pattern type: `f` for fuzzy pattern, `r` for fixed-length wildcard pattern, and `i` for interval-length wildcard pattern. After the pattern type, parameters can be specified in the form of `k = v` (i.e., some parameter `k` is set to `v`). Usually, the parameters are hard-coded values, like adapter sequences and pattern lengths, but they can also reference other, already matched patterns. Since the template file has exactly four lines, it will match every four lines in the input file. Each line of the template file will attempt to match its corresponding line in the input file by checking if the patterns appear, and whether they appear in the correct specified order.

We use multiple interval-length wildcard patterns without any parameters (`{i}`) to match any number of any characters. Note that it is possible to restrict the set of acceptable characters by specifying something like `pattern = "a-z0-9"`.

We also use a fuzzy pattern for the barcode. It pulls the actual barcodes from another list file named `b.txt` (the f before the quotes symbolizes that it is a file path), and allows up to one edit between each barcode and the DNA sequence. It is a required pattern, which means that if it is not matched, then the entire read is not matched by the template file. Also, the barcode is trimmed from the actual DNA sequence once it is matched. Variables that allow the template file to refer to the statistics of the fuzzy pattern match are created using the name `b`.

To ensure that trimming the barcode also trims the corresponding quality scores, we use a fixed-length wildcard pattern that can match any character. The length of the pattern references the variable (which resides in `%...%`) that contains the length of the barcode match, which means that the same number of characters will be removed from both the DNA sequence and the quality score sequence when they are trimmed.

We number the second and fourth lines with increasing indexes to ensure that they are processed by the tool in that order. They are not required for this example because the patterns are matched line by line, but in some cases, it is necessary to match in a certain order so variables get created before they are used.

A comprehensive list of available options and features is available here: https://github.com/Daniel-Liu-c0deb0t/Java-Fuzzy-Search#Fuzzysplit-tool.

The following command is used to run Fuzzysplit demultiplexing:

```
java -jar fuzzysplit.jar \
    input.fastq \
    --pattern template.txt \
    --matched matched_%b.pattern_name%.fastq \
    --unmatched unmatched.fastq
```

This matches reads from `input.fastq` using `template.txt`. Reads that match a certain barcode are trimmed and put into a specific file with that barcode's name. Reads that do not match any barcodes are not trimmed and put into the unmatched file.

It is easy to extend this to paired-end demultiplexing by duplicating the template file and adjusting some parameters. The template file for forward reads can be

```
{i}
1 {f required, trim, name = "f", edits = 1, pattern = f"f.txt"}{i}
{i}
2 {r trim, length = %f.length%}{i}
```

The template file for reversed reads can be

```
{i}
3 {f required, trim, name = "r", edits = 1,
                    pattern = f"r.txt"[%f.pattern_idx%]}{i}
{i}
4 {r trim, length = %r.length%}{i}
```

This allows the reversed template to match for barcodes that are corresponding to ones defined for the forward reads. Also, the reversed reads are matched *after* forward reads due to the line numbers. This ensures that the reversed template has access to variables created during the forward match. Note that the line marked with 3 and the subsequent line should be on the same line when actually running Fuzzysplit.

## TEMPLATE MATCHING ALGORITHM

The core of Fuzzysplit is the matching algorithm for the template files and the input data files. We first examine how each specific type of pattern is matched and then we introduce the overarching algorithm that ties them all together.

### Fuzzy patterns

As each fuzzy pattern may have multiple subpatterns, multiple fuzzy string searches are required to find the matching locations of all subpatterns in a fuzzy pattern.

Fuzzysplit uses the following well-known recurrence to search for a string subpattern $P$ of length $|P|$ in some text $T$ of length $|T|$ with the Levenshtein distance metric (*Levenshtein, 1966*):

$$d(i, 0) = 0 \tag{1}$$

$$d(0, j) = j \tag{2}$$

$$d(i, j) = \begin{cases} d(i - 1, j - 1), & \text{if } T_i = P_j \\ 1 + \min\{d(i - 1, j - 1), d(i - 1, j), d(i, j - 1)\}, & \text{otherwise} \end{cases} \tag{3}$$

where $d(i, j)$ indicates the Levenshtein edit distance between $P_{1 \dots j}$ and $T_{1 \dots i}$ for a match in $T$ that ends at index $i$ (*Sellers, 1980*). This property allows us to find match locations within a certain number of edits, where substitutions, insertions, and deletions all cost one edit. With dynamic programming, finding all matches runs in $O(|P| \cdot |T|)$ time. Fuzzysplit supports allowing transpositions to cost one edit (*Wagner & Lowrance, 1975*) and allowing custom wildcard characters by using variations on the algorithm above. Furthermore, Fuzzysplit allows partial overlaps between each subpattern and the ends of the text. These options allow the matching algorithm to be fine-tuned for specific use cases.

There are many ways to improve the efficiency of the algorithm above, especially with bit-vectors (*Myers, 1999*; *Hyyrö, Fredriksson & Navarro, 2005*; *Wu & Manber, 1992b*). We use Ukkonen's cutoff heuristic (*Ukkonen, 1985*) to improve the average time complexity to $O(k \cdot |T|)$, where $k$ is the maximum number of edits allowed. We choose this instead of the faster bit-vector techniques in order to retain length information about the matches. This technique is also used by Cutadapt (*Martin, 2011*).

For fuzzy searching with Hamming distance (*Hamming, 1950*), Fuzzysplit uses the Bitap algorithm (*Baeza-Yates & Gonnet, 1989*; *Wu & Manber, 1992b*) to allow matching with the Hamming distance metric. It has a worst-case time complexity of $O(k \cdot |T| \cdot \lceil |P|/w \rceil)$, where $w$ is the length of a computer word. We use 63-bit words in Fuzzysplit.

We adopt the use of $n$-grams to quickly eliminate subpatterns before attempting one of the more time-consuming algorithms above. Fuzzysplit splits each subpattern into overlapping $n$-grams. If the text does not contain any $n$-grams of a specific subpattern, then we do not need to match that subpattern. However, for shorter subpatterns, edits within the text may cause none of the $n$-grams to match. Therefore, we only use the $n$-gram method to eliminate a subpattern $P$ if its length satisfies the following inequality:

$$|P| > (k+1)(n-1) + k \tag{4}$$

where $n$ is the $n$-gram size. The inequality represents the worse-case scenario where the edits are evenly spaced in the text and they split the text into segments that are just one short of the $n$-gram size. It is easy to see that if the inequality is satisfied for a subpattern, then it must have at least one $n$-gram that matches with a $n$-gram from the text if they are supposed to match within an edit distance of $k$ by the *Hamming (1950)* or *Levenshtein (1966)* metric. This equation only covers Hamming and Levenshtein distance, but extending it to consider transpositions is trivial.

By default, Fuzzysplit automatically chooses the maximum $n$-gram size possible for all subpatterns in order to maximize the number of subpatterns eliminated. This is because the frequency of each unique $n$-gram is non-increasing as the $n$-gram size gets longer. For a subpattern $P$, the maximum $n$-gram size possible can be obtained by solving the inequality above for $n$:

$$n < \frac{|P| - k}{k + 1} + 1 \tag{5}$$

We use the minimum of all such maximum $n$-gram sizes in order to satisfy the $n$-gram size constraint for all subpatterns. In Fuzzysplit, $n$-grams should not be used if there are wildcard characters within the subpatterns or the text, as they are not considered when generating $n$-grams.

## Fixed-length wildcard patterns

Searching for a fixed-length wildcard pattern in some text $T$ is trivial in $O(|T|)$ time by simulating a sliding window the length of the pattern and checking if the characters in the window are all part of the pattern.

## Matching multiple interval-length wildcard patterns

Unlike matching fixed-length wildcard patterns and fuzzy patterns, we match multiple consecutive interval-length wildcard patterns at the same time. For a list of interval-length wildcard patterns $\mathscr{P}$ of length $|\mathscr{P}|$, we wish to match each pattern $\mathscr{P}_i \in \mathscr{P}$ to contiguous, non-overlapping segments of the text $T$, in order. Each interval-length wildcard pattern $\mathscr{P}_j$ only matches some segment of text $T_{a \ldots b}$ if $\min_j \leq b - a + 1 \leq \max_j$, and every character in $T_{a \ldots b}$ is present in the set of allowed characters for $\mathscr{P}_j$ (i.e., $\forall i \in \{a \ldots b\}$, the character $T_i \in A_j$, where $A_j$ is the set of allowed characters of $\mathscr{P}_j$). For example, the $j$-th interval-length wildcard pattern represented by "A-Z", where $\min_j = 0$ and $\max_j = 3$, matches "ABC" and ""(empty sequence), but not "123" and "ABCD", as it matches any sequence of uppercase letters of some length between zero and three.

We formulate the solution to the problem of matching multiple interval-length wildcard patterns $\mathscr{P}$ with a segment of text $T$ as a recursive function. We define the recursive function as $f(i, j)$, which returns a nonzero value iff $\mathscr{P}_{1 \ldots j}$ matches some $T_{1 \ldots i}$:

$$f(0, 0) = 1 \tag{6}$$

$$f(i, j) = \sum_{k=i-\max_j}^{i-\min_j} f(k, j-1) \cdot [T_c \in A_j, \ \forall c \in \{k+1 \ldots i\}] \tag{7}$$

where $[F]$ evaluates to 1 for some expression $F$ if $F$ is true. Otherwise, it evaluates to 0. The answer to whether the entire $\mathscr{P}_{1 \ldots |\mathscr{P}|}$ matches the entire $T_{1 \ldots |T|}$ is given by whether $f(|T|, |\mathscr{P}|) \neq 0$.

The intuition behind the algorithm is that it attempts to continue a match for a previous pattern by matching the current pattern right after the previous pattern's match location in the text. A continuation is only possible if there is a valid number of characters in the text after the previous pattern and all of those characters are allowed by the current pattern. More formally, for each $i \in \{1 \ldots |T|\}$ and $j \in \{1 \ldots |\mathscr{P}|\}$, we assume that the current pattern $\mathcal{P}_j$ must end at $i$ in the text. That implies that $\mathcal{P}_j$ must start at some index $k + 1$ where $i - \max_j \leq k \leq i - \min_j$ in order to satisfy the length restriction for interval-length patterns. Furthermore, each character in $T_{k+1 \ldots i}$ must be allowed by the current pattern $\mathscr{P}_j$. This whole transition relies on calculating the answers for the previous pattern $\mathscr{P}_{j-1}$ that ends at different indexes in the text, which is essentially solving the exact same problem. This eventually leads to the base case where an empty list of interval-length patterns matches an empty piece of text.

A naive implementation using dynamic programming to cache intermediate function calls results in a loose upper bound of $O(|T|^3 \cdot |\mathscr{P}|)$ for the time complexity, which is very inefficient. We improve the time complexity by using two methods:

- We can speed up the check for whether a pattern allows all of the characters in a segment of text by precomputing the longest length of contiguous characters that ends at a certain index and only has characters that are allowed by a certain

pattern. We show this value for each interval-length pattern $\mathscr{P}_j$ and ending index $i$ as $l(i, j)$, where

$$l(0, 0 \ldots |\mathscr{P}|) = 0 \tag{8}$$

$$l(i, \ j) = \begin{cases} l(i - 1, \ j) + 1, \ \text{if} \ \mathscr{P}_j \ \text{allows} \ T_i \\ 0, \ \text{otherwise} \end{cases} \tag{9}$$

- We notice that the overlap between the segment in the text where the current pattern $\mathscr{P}_j$ has to start its match ($[i - \max_j + 1, i - \min_j + 1]$) and the segment where the text characters are all allowed by $\mathscr{P}_j$ ($[i - l(i, j) + 1, i]$) is the only region in the text where $\mathscr{P}_j$ can begin, if it must end at index $i$. That means that we only need to query a consecutive region for whether the previous pattern that ends anywhere in that region matches the text. For this task, we keep separate prefix sums of $f(i, j)$ for each interval-length pattern:

$$p(0, 0) = 1 \tag{10}$$

$$p(i, \ j) = p(i - 1, \ j) + f(i, \ j) \tag{11}$$

Finding the sum of some segment $f(a \ \ldots \ b, j)$ for some pattern $\mathscr{P}_j$ becomes $p(b, j) - p(a - 1, j)$, an $O(1)$ time operation.

The optimizations above directly lead to the following recursive solution:

$$f(i, j) = p(i - \min_j, j - 1) - p(\max\{i - \max_j, i - l(i, j)\} - 1, j - 1) \tag{12}$$

We implement this in Fuzzysplit for a total run time complexity of $O(|\mathscr{P}| \cdot |T|)$ by using $|\mathscr{P}| \times |T|$ dynamic programming matrices for $l$, $p$, and $f$.

## Overarching algorithm

Fuzzysplit uses a greedy overarching algorithm that matches a list of arbitrary patterns $\mathscr{P}_{1 \ldots |\mathscr{P}|}$ from one line of the template file with its corresponding line of input text $T_{1 \ \ldots \ |T|}$. First, it partitions the list of patterns into continuous chunks of either fixed-length patterns (both fuzzy patterns and fixed-length wildcard patterns) or interval-length patterns. The result is adjacent chunks with alternating general pattern types. The general idea of the overarching algorithm is to search for candidate match locations for each entire chunk of fixed-length patterns and checking whether the interval-length patterns immediately before that chunk matches the text. We show a sketch of the algorithm in Algorithm 1. Note that the first and last patterns must be separately handled to take into account start and the end of the line of input text in the actual implementation.

To search for a matching location with an entire fixed-length pattern chunk, Fuzzysplit goes through each pattern in the chunk and uses each of them as the starting pattern until there is no more patterns left or if a valid match is found. The starting pattern is first searched by itself throughout the input text. For each match index of the starting pattern, Fuzzysplit first attempts to match the interval-length chunk before the current fixed-length chunk, and then it matches rest of the patterns in the fixed-length chunk that are after the starting index. If valid matches are found for both the interval-length chunk and the

---

**Algorithm 1** Matching one line of input text with its corresponding patterns from the template file.

1: $start \leftarrow 0$

2: **for** interval-length chunk $\mathscr{P}_{i...j-1}$ and fixed-length chunk $\mathscr{P}_{j...k} \in$ all chunks **do**

3:     **for** $s \in \{j \ldots k \mid \mathscr{P}_{j...s-1}$ **are all not required**$\}$ **do**

4:        $loc \leftarrow$ list of candidate match locations of $\mathscr{P}_s$ in $T_{start...|T|}$

5:        **for** $m_{lo}$ and $m_{hi} \in loc$ **do**

6:           $end \leftarrow$ end location in $T$ when matching $\mathscr{P}_{s+1...k}$ against the prefix of $T_{m_{hi}...|T|}$

7:           **if** $end$ exists and $\mathscr{P}_{i...j-1}$ matches $T_{start...m_{lo}}$ **then**

8:              $start \leftarrow end + 1$

9:              continue outer-most loop and handle the next pair of chunks

10:           **end if**

11:        **end for**

12:     **end for**

13:     **return** required pattern not matched match found

14: **end for**

15: **return** match found

---

fixed-length chunk, then Fuzzysplit repeats the same algorithm with the next pair of interval-length and fixed-length pattern chunks. Note that if all the patterns in a fixed-length chunk are not required and none of the patterns actually match the text, then the interval-length chunks before and after the fixed-length chunk are merged into a larger interval-length chunk.

Unfortunately, the algorithm we describe is not efficient, as the matching operation for the interval-length chunk and the fixed-length chunk is repeated for each starting pattern and for each match location. We optimize the algorithm with two simple strategies. First, we cache the result of matching each fixed-length pattern at some location in the input text so they can be accessed in $O(1)$ time without having to be matched again at that location. Then, we cache the dynamic programming matrix used for matching interval-length patterns, which can be extended without recalculating previously calculated dynamic programming values. This is possible because the *start* location in the input text to start matching the interval-length chunk is always the same no matter where the current fixed-length chunk matches, as the algorithm never backtracks on previous chunk matches. Only the end location of the interval-length pattern changes as different candidate match locations of the current fixed-length chunk are examined. Overall, the worst-case amortized time complexity for matching the interval-length chunk across all of the candidate match locations of the current fixed-length chunk is linear to the length of the input text.

The algorithm we describe greedily picks the first encountered valid match for the current interval-length and fixed-length chunks that is valid. This is efficient, as it can terminate early if a required pattern is not matched, but it does not lead to the most optimal match. In some cases, previous greedy matches of fixed-length chunks may

cause later chunks to run out of space in the text. We choose to implement this greedy algorithm instead of an backtracking exhaustive search algorithm that finds the best possible match due to time complexity concerns.

We also use a greedy approach for matching within a chunk of fixed-length patterns after a starting pattern match candidate has been found for the chunk. The fixed-length patterns are matched one by one, without any backtracking. For fuzzy patterns, Fuzzysplit greedily selects the match with the lowest edit distance.

## Multithreading

As reads can be handled independently of each other, parallelizing the matching process is very feasible with the producer-consumer paradigm. Fuzzysplit uses the main thread to read in batches of reads from all input files and place them in a queue. Meanwhile, it spawns multiple worker threads that handle batches of reads from the queue in parallel. Each thread handles one batch of reads (each read is handled independently of other reads, even within a batch) and then outputs them as soon as it is done. The output step is synchronized to allow only one thread to write out to some file at a time. Batching reads allow a reduction in output synchronization at the expense of increase memory usage. Furthermore, the main thread may block to wait for the worker threads by using a semaphore, in order to constrain the amount of reads stored in memory at one time.

## RESULTS

### Adapter trimming

We test Fuzzysplit on the simple task of searching for 3′ adapters. The input data is one FASTQ file with 10 million reads. Around 75% of the reads have a randomly generated 3′ adapter that is 30 nucleotides long. Each read's adapter has a 50% chance of having no edits, and 50% chance of having one edit. In total, each DNA sequence is padded with random nucleotides to around 130 nucleotides long, with or without the adapter. We generate two sets of data, one with substitution edits only, and another with insertions, deletions, and substitutions. The data simulation scripts are available here: https://github.com/Daniel-Liu-c0deb0t/Java-Fuzzy-Search/tree/master/scripts.

We measure the run times for a few different settings and we show them in Fig. 1. Each run time is relative to the baseline at 100%. The baseline is mostly the default settings in Fuzzysplit: allowing up to one edit based on the *Levenshtein (1966)* metric, using a single thread, and using the default adaptable *n*-gram setting (in this case, it uses an *n*-gram size of 15 due to the adapter length being 30 nucleotides long). Each setting shown in Fig. 1 is a modification relative to the baseline. Also, all tests were ran on the data set with insertions, deletions, and substitutions except for the run that uses the *Hamming (1950)* metric. Note that Fuzzysplit perfectly found all reads with adapters.

We find that increasing *n*-gram sizes does indeed lower the run time, as it is better at eliminating subpatterns before the *Levenshtein (1966)* edit distance is calculated. Running the program with two threads does improve the run time. We do not show more than two threads because the run time gets worse due the testing machine being limited to only two CPU cores. However, we expect the run time to continue to improve with
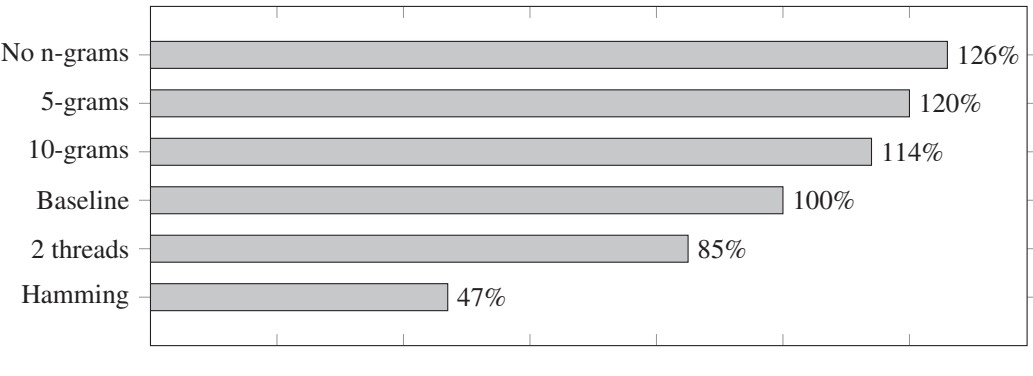

**Figure 1 Comparison of adapter trimming run times of different settings relative to the single-threaded baseline (100%) that uses 15-grams for filtering and computes Levenshtein distance.** Lower is better. The overall run time of the baseline is approximately 10 min. All parameter variants are relative to the baseline. E.g., 10-grams means that matches are filtered with 10-grams instead of the default 15-grams in this case.

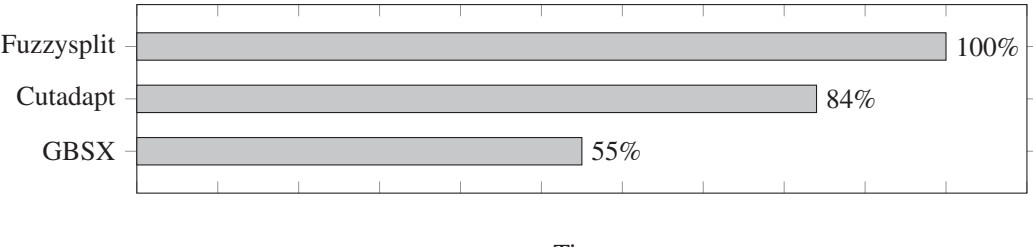

**Figure 2 Comparison of demultiplexing run times of other tools relative to Fuzzysplit (100%).** Lower is better. Fuzzysplit takes approximately 10 min to run.

multithreading if more cores are available. Finally, as expected, matching with the *Hamming (1950)* metric drastically improves the run time.

## Demultiplexing

We first generate 48 random barcodes of varying length between 8 and 15 nucleotides long. Then we generate 10 million reads, where each read's DNA sequence has a 50% chance of having one of the 48 5′ barcodes. Overall, each DNA sequence consists of a possible barcode sequence and 100 random nucleotides. Each barcode can have up to one insertion, deletion, or substitution edit, similar to how the adapters were generated.

We test the run time for Fuzzysplit, Cutadapt (*Martin, 2011*), and GBSX (*Herten et al., 2015*), and we show the results in Fig. 2. Fuzzysplit was ran using two threads, the default *n*-gram setting, and allowing one edit with the Levenshtein metric. For Cutadapt (*Martin, 2011*), each barcode was anchored to the 5′ end and we used 0.125 as the edit threshold for Levenshtein distance. GBSX (*Herten et al., 2015*) was also ran with allowing one edit with *Levenshtein (1966)* distance, but checking for enzymes and 3′ adapters were disabled to match the other tools. Overall, Fuzzysplit is slower than the other tools since it trades speed for much greater flexibility.

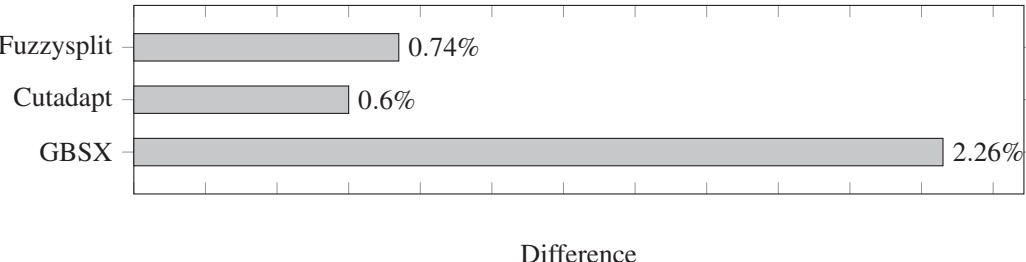

**Figure 3** **Demultiplexing difference percentages of different tools.** Lower is better.

The difference percentages for all three tools are shown in Fig. 3, where each difference $D$ is defined as

$$D = \frac{1}{\text{with\_barcode}} \sum_b \text{wrong}_b + \text{missing}_b \qquad (13)$$

The difference metric counts both the number of reads that are missing from a file and the number of reads that are not supposed to be in that file, and divides the sum by the total number of reads that have a barcode. Overall, the difference percentage of Fuzzysplit is comparable to that of Cutadapt (*Martin, 2011*), and GBSX (*Herten et al., 2015*) is less accurate than either tools. Note that Cutadapt and Fuzzysplit both resulted in more false positives than false negatives over all barcodes, while the opposite was true for GBSX.

## LIMITATIONS

The flexibility of Fuzzysplit comes at a steeper learning curve for using the tool, since custom template files must be written for each format. Also, error messages become less user-friendly due to the tool being more generalized. Furthermore, binary file formats like BAM cannot be directly processed, since the tool can only match patterns in plain text formats. The tool is also unable to interpret quality scores in FASTQ formats and perform quality trimming or filtering, because they do not involve matching patterns within the input text, and they are specific to certain input formats.

Also, Fuzzysplit's method of splitting the input data with a delimiter and matching each splitted portion of the input with its corresponding line in the template file results in its inability to easily handle the FASTA format that allows the DNA sequence to fill an arbitrary number of lines. For future work, this issue can be addressed by allowing greedy pattern matches that possibly span multiple lines.

## CONCLUSION

We created a new flexible pattern matching tool that can be used for trimming and demultiplexing. Overall, we find that it is comparable to existing tools in terms of matching accuracy. However, it is much more flexible than previous tools due to how it uses a simple declarative language to represent how each pattern should be matched.

### Funding
The author received no funding for this work.

### Competing Interests
The author declares that he has no competing interests.

### Author Contributions
- Daniel Liu conceived and designed the experiments, performed the experiments, analyzed the data, contributed reagents/materials/analysis tools, prepared figures and/or tables, authored or reviewed drafts of the paper, approved the final draft.

### Data Availability
Fuzzysplit code is available at GitHub: https://github.com/Daniel-Liu-c0deb0t/Java-Fuzzy-Search.

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
