# Peer review of "Fuzzysplit: demultiplexing and trimming sequenced DNA with a declarative language"

_PeerJ, doi:10.7717/peerj.7170_

## Round 0.1 · original submission · Minor Revisions

Both reviewers suggest, and I agree, to place more emphasis on the generality of the DSL approach. Normally, this would be considered a major revision. However, we all agree that the main application of your approach, as presented, is done well, and simply hinting at other possibilities may be enough in this case. I suggest you carefully weigh each comment and decide which ones can be fully addressed and which ones partially.

·

Basic reporting

The paper is generally well written, and presents an interesting approach to the problem area. That said, I have some suggestions for improvements:

1 - In the FuzzySplit section, the 'Interval-length wildcard patterns' concept, needs a little more detail, in particular why such a concept is needed.

2- There is no real description of the "template language" in the manuscript, beyond the subset which is used in the Example section. This could either be done as an extra section before or after the example.

3 - At the end of the Example section, it briefly mentions paired-end support as being 'easy' - I would not expect most readers to have sufficient understanding of the tool to achieve this, and thus would suggest expanding this section to cover this.

4 - I'm somewhat confused by the multithreading explanation - given that the read structure seems to depend on the template file, how is it possible to 'batch reads' before matching them against the template?

5 - Missing 'were' in the last line of the Result section, Demultiplexing paragraph.

6 - The figure legends are missing information to help interpret the figures (it's in the main text nearby, but the legends themselves should be more informative). For Figure 1, the baseline should also be plotted, and the figure legend describe the key parameters (threads, ngrams etc).

7 - The figures lack x-axis values, and the provided percentages, for figures 1 and 2, are relative values and thus not useful for estimating how long another data set may require to process.

8 - In the case of figure 3, it would be useful to know whether the errors were false positives or false negatives. Given the random sequence generation approach, one would expect some false positives, but false negatives would be more concerning

Experimental design

It would be beneficial to have some concrete example scenarios which motivated the high-flexibility approach chosen.

Validity of the findings

No comment

·

Basic reporting

The English is clear. There is some ambiguity in the mathematical notation that should be fixed to improve clarity. Some nonstandard terminology was used.

The section “Matching Multiple Interval-Length Wildcard Patterns” is hard to follow. It should clearly state what the problem is and what the used mathematical symbols mean. Perhaps adding a drawing would be a good idea?

Data availability: Evaluations were done on simulated data only, which is sufficient for this type of problem, but the scripts should be made available. It appears as if they are in the GitHub repository; this could just be mentioned in the manuscript.

The program is missing a license. I would suggest an Open-Source license to match the spirit of PeerJ. The chosen license should be mentioned in the manuscript.

Experimental design

The introduction aims to advertise usage of a new domain-specific language for preprocessing of sequencing reads. The major advantages given is the greater flexibility of such an approach. However, the only example given is that of parsing and demultiplexing FASTQ files, which is not compelling enough of itself.

I would suggest to give at least one more example here (best would a problem that other tools cannot solve easily because of their inflexibility, a contrived example would be ok), but also to shift emphasis in the introduction towards the actual knowledge gap this paper addresses, which is (roughly) how to write an efficient interpreter for a pattern-matching domain-specific language (DSL).

Validity of the findings

The idea of using a DSL for sequencing data preprocessing is a nice one, but I would also like to see a discussion of the limitations of the approach. I can imagine some disadvantages:

* Error messages could become cryptic: “Pattern abc incomplete at line xyz” vs. “The FASTQ record is missing quality values"

* Yet another language to learn (a small one, but nonetheless)

* Every user has to re-implement a FASTQ/FASTA parser

* I guess handling SAM files becomes complicated (does it?)

* Binary files (BAM files) cannot be handled (or can they?)

Additional comments

Major comments

1. What is proposed is a so-called “Domain-Specific Language (DSL)” and this should be mentioned somewhere.

2. As stated above, the section “Matching Multiple Interval-Length Wildcard Patterns” is somewhat hard to follow. What actually *is* a wildcard pattern P_j, and what does it mean that “characters are allowed by a pattern”?

For example, with the chosen notation, I would immediately expect that both T and P are strings, but only T actually is. P is some kind of sequence of patterns P_j, but what these individual P_j are is undefined.

Giving examples for the three types of patterns on page 2 would probably already help a lot, but also a mathematical definition is needed.

(With that improvement, it should also become clear whether “fixed-length wildcard patterns” are a special case of “interval-length wildcard patterns”, but it would not hurt to state this explicitly.)

Regarding “the formula assumes”: This anthropomorphism should be removed (the formula itself does not assume).

Second bullet point on the page: The notation {x...y} is not explained.

3. There seems to be a connection to regular expressions. A multiple-length interval pattern appears to be the same as a regular expression with length restrictions such as [a-z]{3,7} (3 to 7 occurrences of one of the characters a, ..., z). If this is correct, I suggest to mention this connection. Optional: Have you compared runtime of your implementation to grep, which can also deal with this type of expression?


Minor comments

* (Introduction) “The searching algorithms must also be fuzzy”: As can be seen from the paper titles in the list of references, the standard term to describe this type of string search would be “approximate pattern matching”, “approximate string matching” or similar. The “fuzzy” adjective is nonstandard, and mentioning the standard term would help to clear any confusion with fuzzy logic (only reading the title, this was my first thought).

* Paragraph: “We aim to streamline the development ...” Here, you give as reason for using a DSL that the tool itself does not need to be changed if requirements change. But the actual reason for using a DSL is that it is more expressive than a general-purpose programming language. Otherwise, one would not gain anything because modification of some kind of source code is still necessary. Also, in a language that is already very expressive itself (I state this coming from Python), the advantages of a DSL may not be as clear-cut as implied here!

* (page 4, fuzzy patterns) “As each fuzzy pattern may have multiple subpatterns, ...”: Using the term “subpattern” here and subsequently was confusing to me as I would have initially expected a subpattern to be a part of a pattern, like a substring is part of a string (for example, the pattern contains 20 characters, but you can split it up into two 10-character subpatterns). Why not use the term “pattern” only? Then also all the subsequent descriptions are in line with terminology in the literature, a “fuzzy pattern” could then be defined to be a set of patterns. And the \rho notation could be replaced with P, as one would expect.

* “Fuzzysplit splits each subpattern into location-agnostic n-grams.” Are the n-grams overlapping or non-overlapping? Also, an n-gram by itself is location-agnostic by default – no need to mention this.

* How would the DSL handle FASTA files, where the sequence is split over multiple lines? How to do this is not obvious from the given example.


Software suggestion

* To simplify usage, you could create a wrapper script so that the program can be run directly with 'fuzzysplit input.fastq ...' instead of 'java -jar fuzzysplit.jar input.fastq ...'. That the tool is written in Java could be considered an implementation detail and as a user, I shouldn’t have to care about it.


Spelling, grammar, style

* fasta or fastq formats -> please use uppercase “FASTA” and “FASTQ” (multiple occurrences)

* demultiplexing reads by a barcodes/indexes -> remove the "a"

* an required pattern -> a required pattern

* we use a fixed-length wildcard pattern can match any character -> ... pattern that can match ...

* the matching locations of all subpattern -> ... of all subpatterns

* Ukonnen’s cutoff heuristic -> Ukkonen

* as a recursion function -> as a recursive function

* GBSX ... is much less accurate than either tools: I suggest to tone this down by removing "much"

* evaluates to 1 for some expression F iff F is true: Use a normal “if”. The next sentence already specifies what happens when F is false.

* Some incorrect lowercase occurs in the references: gbs -> GBS, Gbsx -> GBSX, illumina -> Illumina, BMC bioinformatics -> BMC Bioinformatics, etc. They get lowercased by the BibTeX style. To prevent this, ensure that you enclose the words you want to protect in curly braces (in your .bib file). For example: “A robust, simple genotyping-by-sequencing ({GBS}) approach for high diversity species”

---

## Round 0.2 · Minor Revisions

As you can see from the reviewers' comments, who have gone to great lengths to provide helpful advice and concrete suggestions, there are still several improvements that should be made to the manuscript before publication.

In my opinion, after these are done, this work will present a very interesting approach in an easily accessible way.

·

Basic reporting

Most of my concerns from the initial review seem to have been addressed.

I am still concerned by one point however, regarding the identification of individual reads when using multi-threading. Given that the template seems to define the record structure (e.g. 4 lines of template, matching the length of a FASTQ record), how can the boundaries of each read be found without applying (to at least some extent) the template in the 'main' thread? This seems even more difficult if applied to a format with a variable number of lines per record, such as FASTA. On the other hand, if partial use of the template is required during this partitioning process, which is by definition single-threaded, it would seem to represent a real scalability bottleneck.

Experimental design

No comment

Validity of the findings

No comment

·

Basic reporting

Most of my minor concerns have been addressed, but in my opinion, two major ones were not that need some work.

I realize that when I suggested to write something "clearer" in my initial review, it may not have been obvious to the author what I meant. I have taken the liberty here of writing out entire suggested sentences, but this is in no way intended to mean that they should be transferred literally into the manuscript. Instead, I use this as a way of giving very concrete examples of what I mean.

1. From the rebuttal:

>> As stated above, the section “Matching Multiple Interval-Length Wildcard Patterns” is
>> somewhat hard to follow. What actually *is* a wildcard pattern P_j, and what does it mean that
>> “characters are allowed by a pattern”?
> I have clarified this by defining the problem better in the section.
>> Giving examples for the three types of patterns on page 2 would probably already help a lot, but
>> also a mathematical definition is needed.
> I have not added any examples, since I have made the mathematical notation more clear.

I’ll take the definition of a fixed-length wildcard pattern as an example to illustrate what I meant. Here is how one could write it:

A fixed-length wildcard pattern is a tuple (C, n), where C is a set of characters and n is a nonzero length. A fixed-length wildcard pattern \emph{matches} a text T at position i if T_j\in C for j=i,...,i+n-1. An example is ({G, C}, 5), which matches any substring of length 5 that consists only of G or C.

Or with more prose:

A fixed-length wildcard pattern consists of an allowed set of characters and a length. It is said to match a text at position $i$ if the substring of length $n$ of the text starting at position $i$ uses no other characters than the allowed ones.

I’m not asking for these exact definitions to be added to the manuscript, but the ambiguity in the current definition on page 2 needs to be resolved. In particular, the given definition is unclear about whether there is a single set of allowed characters or whether there is a set for each position. I realize that it becomes clearer later on during the description of the algorithm, but the reader should not have to guess.

>> For example, with the chosen notation, I would immediately expect that both T and P are strings,
>> but only T actually is. P is some kind of sequence of patterns P_j, but what these individual P_j
>> are is undefined.
> I have clarified that P_j is an object that represents an interval-length wildcard pattern in the
> implementation.

Another, earlier sentence that you added already defines what P_j is ("Each interval-length wildcard pattern P_j matches only ..."), so this sentence can be removed again. That P_j is implemented as an object is not important here.

Experimental design

2. From the rebuttal:

>> I would suggest to [...] to shift emphasis in the introduction towards the actual knowledge gap this paper addresses, which is (roughly) how to write an >> efficient interpreter for a pattern-matching domain-specific language (DSL).
> I have added another example for paired-end demultiplexing to better explain the DSL aspect of
> the tool. However, I have not included an example that is exclusive to this tool. The main goal is
> to introduce the idea of applying DSLs to demultiplexing and adapter matching.

My suggestion of changing the introduction has not been addressed.

The current introduction focuses on Fuzzysplit as a tool, which sets the expectations too high: A new tool is usually expected to have more features, be faster or use less memory as the existing ones, which isn’t the case here. The manuscript emphasizes the flexibility of the tool, but what the concrete advantages of this are is unclear. This is not a problem – novelty is not a requirement for publication in PeerJ, so the tool not being better than others is fine. However, you do need to address a "knowledge gap". Here, the knowledge gap is not the tool itself, but the idea of using a DSL, which is what the introduction should focus on.

I think just rearranging a couple of sentences would be enough. For example, instead of saying "In this paper, we introduce our novel Fuzzysplit tool, ...", you could say: "In this paper, we introduce the idea of [using a DSL], and also describe the algorithms to efficiently match [certain types of patterns], which we have implemented in our novel Fuzzysplit tool." That is, just change the framing slightly.

Validity of the findings

no comment

Additional comments

Most of my minor concerns have been addressed, but in my opinion, two major ones were not that need some work. These are listed first below.

I realize that when I suggested to write something "clearer" in my initial review, it may not have been obvious to the author what I meant. I have taken the liberty here of writing out entire suggested sentences, but this is in no way intended to mean that they should be transferred literally into the manuscript. Instead, I use this as a way of giving very concrete examples of what I mean.

- Thank you for listing some of the limitations of the approach, but doing so at the end of the conclusion makes the manuscript end on quite a negative note! I suggest to move those to a separate "Limitations" section before the conclusion. And/or you can turn some of the limitations into "Future work" or "Open questions" items to make them sound less negative. (Declaring something as future work does not meany you yourself promise to work on it.)

- It appears Reviewer 1’s point 4 about splitting the input into chunks for multithreading has not been addressed: For splitting, the position at which to split needs to be known. For example, for FASTQ, valid splitting positions would be after every four lines, but with flexible templates, how is this done without fully parsing the file beforehand?

- From the rebuttal:

>>The notation {x...y} is not explained.
>It indicates an interval.

I have not seen this before. Why not use the standard [x, y] notation? No special explanation would then be necessary.

- "Fuzzysplit splits each subpattern into location-agnostic n-grams that may overlap." Adding the "may overlap" in this revision made the sentence less clear because "may" can be understood to mean that they sometimes overlap and sometimes don’t (which I believe is not correct). Why not simply: "Fuzzysplit splits each subpattern into overlapping n-grams." And as I stated previously, n-grams are "location-agnostic" by nature since they are just strings of length n. If you leave this in, readers like me may start to wonder whether your idea of what an n-gram is is different from theirs.

---

## Round 0.3 · accepted · Accept

The use of a DSL to parse and edit/distribute FASTQ files is a good one, and the presented implementation is of practical value. I still agree with reviewer Marcel Martin that the different types of patterns could be introduced more formally, but since PeerJ is not an algorithmic or mathematical journal, I believe that the level of formalism is sufficient.

#